# Current Progress and Future Directions for Tau-Based Fluid Biomarker Diagnostics in Alzheimer’s Disease

**DOI:** 10.3390/ijms21228673

**Published:** 2020-11-17

**Authors:** Mohammad Arastoo, Richard Lofthouse, Lewis K. Penny, Charles R. Harrington, Andy Porter, Claude M. Wischik, Soumya Palliyil

**Affiliations:** 1Institute of Medical Sciences, University of Aberdeen, Aberdeen AB25 2ZP, UK; mohammad.arastoo@abdn.ac.uk (M.A.); r03rl17@abdn.ac.uk (R.L.); l.penny@abdn.ac.uk (L.K.P.); c.harrington@abdn.ac.uk (C.R.H.); a.porter@abdn.ac.uk (A.P.); soumya.palliyil@abdn.ac.uk (S.P.); 2Scottish Biologics Facility, University of Aberdeen, Aberdeen AB25 2ZP, UK; 3Genting TauRx Diagnostic Centre Sdn. Bhd., Aberdeen AB24 5RP, UK; 4TauRx Therapeutics Ltd., Aberdeen AB24 5RP, UK

**Keywords:** Alzheimer’s disease, Tau, biomarkers, diagnostics, cerebrospinal fluid, blood

## Abstract

Despite continued efforts, there remain no disease-modifying drugs approved by the United States Food and Drug Administration (FDA) or European Medicines Agency (EMA) to combat the global epidemic of Alzheimer’s disease. Currently approved medicines are unable to delay disease progression and are limited to symptomatic treatment. It is well established that the pathophysiology of this disease remains clinically silent for decades prior to symptomatic clinical decline. Identifying those at risk of disease progression could allow for effective treatment whilst the therapeutic window remains open for preservation of quality of life. This review aims to evaluate critically the current advances in the interpretation of tau-based biomarkers and their use to provide insights into the onset and progression of Alzheimer’s disease, whilst highlighting important future directions for the field. This review emphasises the need for a more comprehensive analysis and interrogation of tau within biological fluids, to aid in obtaining a disease specific molecular signature for each stage of Alzheimer’s disease. Success in achieving this could provide essential utility for presymptomatic patient selection for clinical trials, monitoring disease progression, and evaluating disease modifying therapies.

## 1. Introduction

Alzheimer’s disease (AD) is a chronic and debilitating neurodegenerative disease that is characterised and diagnosed clinically by progressive cognitive decline. A dementia diagnosis is made, somewhere globally, once every 3 s and there are 50 million dementia sufferers worldwide. This total is expected to triple by 2050 and approximately two-thirds of these cases are expected to be AD [1]. This devastating disease is straining healthcare systems worldwide at a cost of $1 trillion per year with informal care estimated at 82 billion hours per year [1]. Dementia now kills more people in the U.S. than breast cancer and prostate cancer combined and is now the top cause of death in both England and Wales [1]. AD is now considered the most feared disease by the American public [2].

There is an urgent need to discover, develop and approve disease-modifying drugs for AD to combat the relentless and all-consuming global economic and societal burden. Unfortunately, despite the urgency of the need there has been no success to date. The costly attrition rate of 99.6% of AD drugs in clinical development between 2002–2012 and over 400 failed clinical trials since the last symptomatic AD drug was approved by the United States Food and Drug Administration (FDA), truly highlights the sense of frustration of this research area [3,4]. This trend continues with no disease modifying drugs for AD approved by the FDA or European Medicines Agency (EMA).

This lack of success is due to multiple factors, but one that is gaining increasing attention is the fear that treatment is initiated only after irreversible damage has already occurred. It has been known for some time that the pathophysiological changes of AD begin several decades prior to symptoms [5,6], yet clinical trials to date have focussed on the symptomatic phase of the disease and have been unable to address the extended early window available for therapeutic intervention [5]. However, the long prodromal phase of the illness makes it impossible to rely on conventional clinical endpoints in clinical trials and requires the identification and validation of suitable biomarkers as potential surrogate endpoints. To be useful, there must be a high degree of confidence that such surrogate endpoints are reliable predictors of either continuing progression or successful prevention. These requirements are onerous, although this has been achieved using blood-based biomarkers which permit disease monitoring in, for example, diabetes (glucose) and cardiovascular disease (cholesterol). We argue here that tau-based biomarkers will likely be an important part of a multi-modal biological signature that will permit the various stages in disease progression to be mapped. The tau component of this signature will need to go beyond measurement of levels of total or phosphorylated tau to permit interrogation of a complex and diverse family of protein abnormalities and conformations in blood.

### Evolution towards a Biomarker-Based Diagnostic Framework for Alzheimer’s Disease

Until recently, a diagnosis of probable AD was based largely on clinical history and psychometric testing. These document memory impairment and deficits in a range of cognitive functions, with a requirement for disease pathology at post-mortem to confirm a definite diagnosis [7]. This early diagnostic framework was revised by the National Institute of Ageing-Alzheimer’s Association (NIA-AA) in 2011 with two notable updates. The first was the introduction of biomarker testing to support clinical diagnostic criteria by including analysis of cerebrospinal fluid (CSF) amyloid β and tau. The second was the recognition that the development of AD is a continuum that could be defined as three clear stages of disease (as described in [8] and Figure 1):
Preclinical Alzheimer’s disease [8]
○A disease stage based entirely on biomarker-based changes○No clinical symptoms (presymptomatic)○Currently offers no diagnostic utility for cliniciansMild Cognitive Impairment (MCI) due to Alzheimer’s disease [9]
○Established criteria for the prodromal symptomatic phase ○AD biomarker evidence may be used to support diagnosisDementia due to Alzheimer’s disease [10]
○Refined clinical criteria for the demented phases of AD ○Biomarker evidence may be used to support diagnosis

Although diagnosis is still largely reliant on clinical evaluation, these were important steps on the way to the formal recognition of biomarker measurement of the disease to support a diagnostic framework. Important progress is still being made with redefining Alzheimer’s disease from one determined solely from an individual’s cognitive status to one determined by a disease-specific biological signature. Recent progress towards this goal is the NIA-AA research framework proposed in 2018, which advocates an updated definition of the disease based on in vivo amyloid beta (Aβ), tau and neurodegeneration (ATN) biomarkers (Table 1), even in the absence of clinical symptoms [11]. This framework allows for eight different ATN “biomarker profiles”. How these have been termed and how they relate to the AD continuum and clinical phases of the disease is summarised in Table 2 and Table 3, and Figure 1. 

This effort to provide an unbiased biomarker classification system for AD continues to be regarded as a “work in progress” and should be approached with caution. An important limitation of this framework is its critical reliance on the presence of amyloid biomarkers in defining presymptomatic and symptomatic individuals as falling within the AD continuum to the exclusion of any other signals. This makes the framework unduly reliant on accepting the Aβ cascade hypothesis as beyond dispute. However, this hypothesis is subject to increasing doubt in the light of the multiplicity of trial failures targeting a variety of steps in the proposed cascade [12]. Clear evidence of target engagement in the absence of clinical benefit has led many to rethink the assumptions underlying the hypothesis (Box 1). 

Box 1Brief summary of evidence against the amyloid cascade hypothesis.***AD patients without amyloid pathology:*** Results from a large cohort study of
4000 subjects with mild cognitive impairment (MCI) or AD showed amyloid PET
imaging positivity present in only 54.3 and 70.5 % of cases respectively [13].***Healthy subjects with amyloid pathology***: Neuropathological and PET imaging studies have shown evidence of extensive amyloid pathology in cognitively normal older people that is comparable with AD patients [14,15,16]. Another study has shown a 76 % overlap in Aβ pathology between advanced AD cases and healthy controls [17].***Amyloid pathology: wrong place, wrong time in AD:*** Neuronal loss in AD initially occurs in the entorhinal cortex and hippocampus causing subsequent and characteristic memory loss of AD whereas amyloid plaques are first found in the frontal regions or basal ganglia. Amyloid plaques are not adjacent to this neuronal loss nor is the amount related to the extent of cognitive decline in AD [18,19].***Clear target engagement without disease modification:*** A single dose of verubecestat (β-secretase 1 inhibitor) has been shown to reduce Aβ by up to 90 % in human CSF yet this did not change the trajectory of the disease in clinical trials [20,21]. PET scanning showed significant target engagement in two phase 3 Aβ immunotherapy clinical trials; no clinical improvement was evident [22].

In parallel with these findings, tau has subsequently received increased interest not only as a disease modifying target for AD but also as a potential diagnostic biomarker for AD. This is primarily due to strong correlation between pathologic tau deposition in the brain and the cognitive decline of AD patients [23,24,25]. This staging of tau deposition also correlates with AD associated neuronal loss in the entorhinal cortex and hippocampus causing the subsequent and characteristic memory loss of AD [26,27,28]. Pathologic tau has been shown to mediate neurodegenerative events of AD at a molecular and cellular level. These include but are not limited to:
Microtubular destabilisation and disrupted axonal transport [29]Dysregulation of intracellular calcium [30]Mitochondrial dysfunction [31]Oxidative stress [32]Damage to the proteasome [33]Promotion of neuroinflammation [34]Degeneration of microglia [35]Synaptic dysfunction and loss [36]Altered neuronal activity [37]Neuronal loss [26]

The development of AD from presymptomatic to symptomatic phases is complex and the research framework, designed as it was around information available at the time of publication, requires continued refinement as new evidence becomes available. Future studies including ATN biomarkers complemented by additional candidate biomarkers could improve the prognostic power of the framework. These potential factors are not limited to, but might include: BACE1, TREM2, YKL-40, IP-10, neurogranin, SNAP-25, synaptotagmin, α-synuclein, TDP-43, ferritin, VILIP-1, and NF-L (reviewed extensively elsewhere [38]). For example, neurofilament light chain levels, which reflect the extent of axonal injury, have shown good and continued evidence of being a strong marker for “N” [39,40]. Regardless of opinion or criticism regarding the suggested framework, it is certainly clear that any biological definition of AD is unlikely to be limited to the change of a single biomarker. It is much more likely that a multi-modal biological signature will emerge for various stages in disease progression. However, research would suggest that tau, a complex protein in the physiology and progression of AD, should not be considered as a single biomarker categorised solely into T-tau (total tau) and P-tau (phosphorylated tau) and that the many different forms of tau (which we refer to as the “tauosome”) could at least contribute to the complex and multi-target panel required for improved diagnostic interrogation of AD onset and progression.

This paper aims to review past and current efforts to use tau as a biomarker for AD and highlights how these studies point the way to a possible tau-based biological signature for each stage of Alzheimer’s disease.

## 2. Diagnostic Utility of Tau within CSF

### 2.1. Total Tau and Truncated Tau

Total tau (T-tau) levels in the CSF can distinguish between healthy individuals, those with MCI and AD dementia patients. An increase in CSF T-tau is also predictive of future conversion to later disease stages [41]. Since there is a high degree of heterogeneity at the protein level between various tauopathies, and since CSF T-tau is elevated following neuronal injury caused by other factors such as traumatic brain injury [42] and acute stroke [43], it remains to be seen whether T-tau alone will be useful in a diagnostic setting [44]. Whilst T-tau is a validated clinical biomarker for the AD continuum, more research is required to unlock fully the true potential of tau as an AD-specific diagnostic biomarker. Currently most CSF T-tau tests, such as the AlzBio3 kit (Innogenetics), are based on detection antibodies around the N-terminal half of the tau molecule. These are unable to distinguish between full-length tau and fragments that lack N- or C-terminal regions [45]. This raises concern as to whether current T-tau tests are missing valuable information hidden within the molecule and whether these measurements are truly representative of “total tau”. One area of ongoing promising research is exploring the utility of tau fragments as biomarkers of the AD continuum.

There is growing evidence that truncated tau fragments are present in the AD brain (for an extensive review see [46]). There is a lack of consensus regarding the specific nature of these fragments and their role in disease pathology. Their identification and characterisation could greatly assist in the differential diagnoses of the tauopathies which are often difficult to distinguish at early disease stages. The core of the paired helical filaments (PHFs) which comprise neurofibrillary tangles is restricted to the repeat domain [47,48,49,50] implying that large portions of the C-terminal half of the molecule may be sequestered. Therefore, it may be important to consider the ratio of N- and C-terminal fragments as potentially indicative of the tangle burden within the brain. It has been shown that the majority of fragments within the CSF of AD patients are a mixture of smaller fragments with only a small proportion measurable by ELISA assays that depend on detection of full-length or even N-terminally intact tau [51]. This points to the need for continued research into the pathways that lead to the formation of these fragments in order to gain a better understanding of whether the generation of any specific fragments is upregulated and whether there are specific associations with disease onset and progression.

The majority of tau degradation and clearance is mediated by the proteasome and autophagic degradation systems [52]. It is likely that upstream post-translational modifications, aggregation and interactions with chaperone proteins govern how tau molecules finally arrive into these clearance systems. Numerous sites of truncation have been identified within the tau protein (Figure 2) but it is not yet understood how these arise or how they relate to the AD continuum. Many truncations arise as part of normal physiology but a better understanding of these degradative pathways in various conditions could give rise to the identification of truly disease specific tau fragments and disease specific AD biomarkers.

In CSF, several potential AD-specific fragments are being investigated and it is hoped that these will add to the refinement and diagnostic utility of T-tau measurements going forward. An N-terminally truncated fragment ending at Lys-224 was found to be specific to neurons, found in NFTs, and upregulated in AD [53]. Furthermore, a fragment ending at residue Asp-368, that is cleaved by asparagine endopeptidase (AEP), was found to be upregulated in AD CSF [54]. Only 3–8% of measurable tau in the sample had a C-terminus at residue Asp-368 but a shift to a higher proportion of this fragment in AD CSF would suggest that the fragmentation pattern may change in disease. Interestingly, levels of AEP were found to be lower in AD patients, which suggests that measuring the expression of various tau clearance mediators may have diagnostic merit. A separate study found the levels of a fragment ending at Gly-314, a C-terminally truncated soluble tau species that is responsible for cognitive impairment in transgenic mouse studies, to be higher in the brains of cognitively impaired individuals compared to normal counterparts [55]. The levels of this fragment were modestly predictive of overall cognitive impairment but interestingly the level of caspase-2, a mediator of proteolytic clearance, was found to be higher in cognitively impaired individuals as well. Several C-terminal fragments have also been found to be enriched in late stage disease [56]. Monomeric tau is preferentially cleaved by calpain-1 at a site within the core region of PHFs but upon oligomerisation, there is a shift in fragmentation pattern to a more C-terminally dominant conformation. These data suggest that a transient and malleable tau clearance mechanism might exist within the brain which may be readily altered in response to various stages of aggregation. It remains to be seen if these changes reflect tau clearance mechanisms which represent an adaptive response to pathology and how they contribute to the pathophysiology of the disease as it progresses in the brain.

From these selected publications, it is clear that numerous truncated fragments of tau exist and are linked to disease progression and severity. A greater understanding of this “tauosome” may provide an untapped wealth of information if we look beyond the current limited focus on T-tau measurements.

### 2.2. P-Tau

Phosphorylation plays an essential role in the physiological function of tau, predominantly by regulating axonal growth, plasticity, and transport through altering the microtubule binding affinity of tau [57,58,59]. There are approximately 85 potential phosphorylation sites spanning the full-length tau molecule, with as many as 49 phosphorylation sites detected in the AD brain (Figure 3).

It was first shown over 30 years ago that tau isolated from crude tangle preparations from AD brains is “abnormally’ phosphorylated” [61]. However, the relationship between phosphorylation of tau and AD is still not fully understood and continues to be controversial. Dispute continues over whether phosphorylation is causative, or casually associated or even a protective response delaying AD pathogenesis [62,63,64]. Different tau phospho-sites have also been postulated as having differing roles in the pathophysiology of AD. For example, some phosphorylated sites inhibit pathologic aggregation whereas others promote the process [65,66]. In vitro, spontaneous aggregation of truncated tau into PHFs does not require phosphorylation [67,68] and phosphorylation of tau inhibits pathological tau-tau binding by a factor of about 20-fold [69].

Although the level of phosphorylated tau represents less than 5% of total PHF-tau [70] neocortical neurofibrillary pathology in AD correlates strongly with hyperphosphorylated tau load as determined by immunohistochemistry performed on post-mortem brain samples [71]. Importantly, phosphorylated tau load in the brain correlates with the concentration of phosphorylated tau in the CSF in vivo. It can therefore act as a reliable biomarker for AD and hence its inclusion as a diagnostic criterion [71].

An extensive systematic review and meta-analysis of 91 studies (7498 AD patients, 5126 controls) revealed a fold change of 1.88 on comparison of CSF P-tau between AD patients and controls [72]. When compared with other validated AD CSF biomarkers (T-tau and Aβ), P-tau shows a much greater specificity relative to both AD and NFT burden. This may be due to P-tau, unlike T-tau, being unaffected by possible co-morbidities such as brain injury or stroke. Additionally, P-tau does not correlate with other potential tauopathies such as frontotemporal lobar degeneration which is difficult to distinguish from AD clinically [73,74]. Importantly, P-tau has been shown to be an indicator of the preclinical and prodromal phases of the AD continuum, with an increase of P-tau concentrations detectable some 15 years prior to appearance of symptoms [75].

### 2.3. Expanding from P-tau181

Despite the potential 85 phosphorylation sites spanning the tau molecule, most studies have focused solely on just one phospho-tau site, P-tau181. Other potential AD biomarkers such as P-tau231 [76] and P-tau199 [77] have been studied but not to the same degree as the “gold standard” P-tau181. Although P-tau181 offers clear evidence of diagnostic utility for AD, exploring other phospho-tau sites could further enhance this utility.

Two recent studies have employed mass spectrometry to explore these potential phosphorylation sites as AD biomarkers [78,79]. Following the analysis of 47 phospho-peptides covering 31 different phosphorylation sites, 11 were found to be upregulated by at least 40% in AD CSF [79]. A further mass spectrometry study revealed that tau from AD CSF was not only hyperphosphorylated specifically at the P-tau181 site, but at many other sites including P-tau111, P-tau217, P-tau231, P-tau205, P-tau208 and P-tau214 [78]. In fact, the most notable change in phosphorylation was not P-tau181 (10.1% vs. 13.3%, control and AD, respectively) but P-tau111, which showed 0.8 % phosphorylation in controls compared with 8.1% in AD CSF. Interestingly, P-tau153 and P-tau175 showed modest phosphorylation (0.8 and 0.1%, respectively) in AD CSF but was undetectable in control CSF [78].

Several recent studies have also concluded that P-tau217 may offer improved diagnostic utility for the AD continuum especially when compared to P-tau181. P-tau217 was increased 6-fold in AD CSF when compared to non-AD samples, whereas this increase was only 1.3-fold for P-tau181. P-tau217 was also deemed a more specific marker than P-tau181 for the preclinical phase of AD as determined by its correlation with Aβ PET scans [80]. A similar study supports this data showing increases in CSF concentrations of P-tau217 in prodromal AD and AD dementia, which was several-fold higher than P-tau181 [81]. In addition, the authors also reported a greater longitudinal increase of CSF concentration and improved correlation with Aβ and tau PET imaging. A further related paper [82] quantified P-tau181, P-tau202, P-tau205 and P-tau217 signatures in CSF, across 35 years of the AD continuum. This study showed that P-tau217 and P-tau181 begin to increase around the same time as amyloid plaque formation is initiated, 21 and 19 years prior to symptom onset, respectively. At the time when neuronal dysfunction (measured using ^18^F-fluorodeoxyglucose-positron emission tomography [FDG-PET]) can be detected approximately 13 years prior to the appearance of clinical symptoms, P-tau205 begins to increase. Lastly, as neurodegeneration begins (as determined by clinical decline, initiation of the symptomatic phase and cortical atrophy shown by MRI), tau-PET ligands begin to detect filamentous tau deposits in tangles, with P-tau217 and P-tau181 gradually decreasing. It was also shown that P-tau202 does not increase throughout the course of the AD continuum [82].

To conclude, P-tau181 does not equate to P-tau and should not be reduced to this in the literature. Moving forward, P-tau could represent a component of a more comprehensive phospho-tau signature that encompasses several disease-specific phospho-sites that could help map the AD continuum through the preclinical, prodromal and symptomatic stages of the disease.

### 2.4. Aggregation and Disease-Specific Post Translational Modifications

The aggregation cascade of natively unfolded tau into insoluble filaments is a defining pathological feature of AD notwithstanding the ATN system. Historically, AD diagnosis has only been confirmed by identification of neurofibrillary tangles at post-mortem. With the success of P-tau based AD biomarkers, which appear to be capable of following PHF formation, together with tau-PET imaging of tau PHF deposits, one logical approach may be to monitor tau aggregation directly as a fluid-phase AD biomarker. Such an approach may have the best chance of diagnosing individuals early in the AD continuum. The detection and quantification of soluble tau oligomers, considered an early intermediate species in the aggregation cascade, could provide an ideal diagnostic target in biological fluids. Soluble tau oligomers are thought to represent the toxic form of tau within neurons with the capacity to spread and potentiate neurodegeneration throughout the brain [83]. Elevated levels of oligomeric tau have been reported in the CSF of AD patients compared to an age matched, non-demented control group. The ratio of oligomeric to total tau was also assessed and found to be highest in moderate to severe AD, followed by mild AD and then controls [36]. This study highlights the concept and diagnostic merit of monitoring tau oligomers in the course of the evolution of the AD continuum. However, there are few tau oligomer-specific antibodies currently available [83,84]. Effort is required to develop a more comprehensive panel of such antibodies. Their subsequent utilisation in larger cross-sectional and longitudinal studies will better ascertain the suitability of oligomeric tau as an early stage AD biomarker.

Post translationally modified tau species may also serve as useful AD biomarkers. Tau is substantially altered through numerous post translational modifications (PTMs). The most common PTM is phosphorylation and it is therefore not surprising that this has received the most research interest thus far. However, tau can undergo numerous other modifications including acetylation, glycosylation, ubiquitination, nitration, methylation, sumoylation, and truncation [59,85]. In stark contrast to the healthy brain, tau is *N*-glycosylated in the AD brain [86,87]. Paired helical filaments from AD brains are glycosylated and whilst glycans have been reported to modulate tau aggregation, aberrant glycosylation may also occur at an early stage of tau pathology and it has been suggested that this precedes abnormal phosphorylation [88]. Similarly, acetylation of tau is increased under pathologic conditions. Because acetylation and ubiquitination can occur at the same lysine residues, acetylation can reduce tau ubiquitination, thereby preventing its proteasomal degradation [89]. In accordance with this, tau acetylation as well as total and phosphorylated tau are elevated in soluble fractions from human brain (frontal cortex) samples at early and at moderate Braak stages of tauopathy compared to brain samples at Braak stage 0 [90]. These studies therefore suggest that abnormally glycosylated and acetylated tau could represent promising biomarkers for early AD diagnosis. Furthermore, different tauopathies have distinct PTMs, which could be utilised to distinguish AD from other tauopathies [91]. A more in-depth understanding of tau PTMs in relation to the pathophysiology of the disease could eventually lead to the addition of PTM-based biomarkers to enrich the current AD biomarker repertoire.

## 3. Translation of CSF AD Biomarkers to the Periphery

CSF biomarkers can give valuable information about the rate of protein production and clearance at a given point in time within the central nervous system (CNS). This has been shown to aid with the clinical diagnosis of other CNS disorders including multiple sclerosis [92], Guillain-Barré syndrome [93], cancers of the brain or spinal cord [94,95], and serious bacterial, fungal, and viral infections [96]. CSF biomarkers used for AD diagnosis have also proven to be a valid proxy for monitoring neuropathological changes in AD.

However, one must consider the scale of the global burden of AD. Imaging and CSF-based diagnostics simply cannot be considered as a large scale and readily deployable primary diagnostic approach for cost-effective population screening. It was estimated in 2015 that around 480 million people worldwide were at Braak stage 2 or later [97]. CSF-based measurements involve potentially dangerous and certainly unpleasant lumbar punctures whilst PET imaging is limited to only the best equipped medical facilities. Attempting to roll out a primary diagnostic screening framework for non-symptomatic and preclinical AD is simply not viable with these current tools. New approaches that allow rapid, low-tech analysis of biomarkers in more readily accessible fluids such as a finger-prick of blood, could be transformative for routine detection of AD at early disease stages. More invasive and hospital-based procedures could then be reserved for use as secondary and confirmatory diagnostic tools if needed. The holy grail of AD biomarker research therefore remains the provision of a cheap, accessible, straight-forward blood test that can reliably diagnose the AD continuum, or at the very least identify those who would require CSF or PET secondary testing. Whilst a home-testing kit for AD remains, in the eyes of most researchers, something of a pipedream, the utilisation of new ultrasensitive techniques such as single molecule array (SIMOA) [98], immunomagnetic reduction (IMR) [99] and next generation mass spectrometry [100] all offer considerable promise as techniques capable of detecting tau biomarkers in a few drops of blood.

### 3.1. From CSF to Plasma: T-tau, P-tau181, and Tau Fragments

The logical diagnostic journey for suitable blood-based AD biomarkers is to pursue the candidates that were validated first in CSF. The evidence for the use of T-tau as a plasma-based AD biomarker has been relatively robust to date, with a systematic review and metanalysis of 271 patients with AD (6 cohorts) and 394 controls (5 cohorts) showing average fold increase of 1.95 for T-tau over controls [72]. Since this analysis, plasma T-tau has been shown to be as strongly associated with AD dementia as CSF T-tau and higher plasma T-tau is associated with faster progression of AD [41,101]. However, the data are not conclusive with some studies showing no statistical difference or a large degree of overlap between control, MCI, and AD cohorts in terms of plasma T-tau levels [38,102].

Three independent studies have shown that P-tau181 in plasma can differentiate between AD, non-AD neurodegenerative disease and control subjects. Plasma P-tau181 was also shown to be increased in the earliest preclinical change of the AD continuum and correlated well with Aβ PET load [103,104,105]. An 8-year longitudinal arm of one study also showed that high plasma concentrations of P-tau181 were associated with cognitively unimpaired and MCI participants who went on to develop AD dementia [103]. A recent large study has shown high correlation between elevated plasma levels of P-tau217 and brain amyloid levels measured by PET [106]. By measuring plasma levels of P-tau217, this study differentiated AD patients from patients with other tauopathies and controls, with a significantly higher diagnostic accuracy than P-tau181, neurofilament light chain and MRI-based biomarkers. In addition, significantly higher levels of plasma P-tau217 were measured in *PSEN1* mutation carriers compared to non-carriers. Participants in this cohort had a mean age of 35.8 and elevated P-tau217 levels were noted in individuals as young as 25 years of age, which amongst mutation carriers is 20 years prior to estimated onset of MCI.

Of the few studies that have investigated the importance and diagnostic value of novel tau fragments in plasma, an N-terminal fragment ending at Ser198 was found to distinguish healthy controls from AD-MCI and AD patients [51]. This study could not distinguish between AD-MCI and AD cohorts suggesting the difference was linked to an early pathological event in the AD continuum that was soon saturated. Interestingly, this study also showed T-tau and N-terminal fragments ending at Lys224 did not differentiate between healthy controls, AD-MCI, and AD patients [51].

### 3.2. Blood-Based Considerations

The reliable interpretation of values for biomarkers present or transferred to CSF and peripheral vascular compartments from deep within the CNS requires new thinking to understand what has been seen to date. The plasma milieu differs substantially from that of the brain and CSF. Brain-derived proteins that do end up in the plasma may undergo a variety of potential alterations prior to eventual proteolytic clearance. The half-life of tau in plasma is only 10 h [98] compared to approximately 20 days in CSF [107]. Tau also exists in a much more dilute state within a matrix of many other potentially confounding proteins such as plasma albumins and circulating endogenous antibodies.

Furthermore, it has been reported that there is a higher proportion of full-length tau in plasma compared to CSF [51] which raises the possibility of a peripheral source of tau that might be a further confounding factor in any diagnostic test. Tau expression has been noted in peripheral tissues such as kidney, liver, testis, and skeletal muscle of rodents [108,109] and tau is also present in human peripheral tissue [110]. One possible explanation for the inconsistencies that have been reported could be the presence of “big tau”. This is a higher molecular weight tau encoding 733 amino acids and contains sequences homologous to 2N4R tau plus a non-homologous stretch of 237 amino acids in the mid-region. Big tau was originally identified in the rodent peripheral nervous system and has since been identified in human peripheral tissues [110,111].

One approach, that avoids these potentially confounding factors, is to confirm that the antibodies used in any test are specific to the CNS form of tau. Another option would involve the isolation of neuronally-derived exosomes that are known to be present in plasma and use these to study AD biomarkers. Several studies have reported that P-tau181 and T-tau derived from plasma exosomes show measurable and significant differences when compared to healthy controls, MCI, and AD, and is more representative of the CSF [112,113].

### 3.3. Extension of AD Biomarkers to Further Peripheral Matrices

Whilst clearly a challenge, blood-based AD biomarkers seem the most logical next step if one were to realise a rapid screening approach at the population level. However, additional periphery-based AD biomarkers are also being considered including saliva. By utilising a combined immunoprecipitation and mass spectrometry approach, the P-tau181/T-tau ratio in saliva was found to increase in a small cohort of AD patients when compared to age-matched controls [114]. A subsequent study using western blotting reported that the P-tau396–404/T-tau ratio was also increased in a small cohort of AD patients when compared to age-matched controls. Despite statistical significance, the data showed a large variation in the values obtained and concluded that salivary levels of P-tau181 could not be used to differentiate between AD and controls [115]. Neither of these studies were able to show a difference in salivary T-tau levels in AD patients compared with controls and this was further supported by a larger study utilising SIMOA-based technology [116].

## 4. Conclusions

The ever-pressing challenge of diagnosing AD at the earliest possible stage of the disease process necessitates the pursuit and discovery of novel biomarkers. As we continue to validate novel biomarkers for AD, we expand our understanding of the molecular mechanisms of AD pathogenesis. This can translate into a more accurate diagnosis of disease stage, particularly in presymptomatic AD, and potentially permit monitoring of progression and response to treatment. The ultimate goal would be to define each stage of the AD continuum using a validated biological signature of AD-specific biomarkers. In doing so, AD treatment could permit a personalised and stratified therapeutic approach that is appropriate for each disease stage. Presymptomatic diagnosis may eventually allow early therapeutic interventions to interrupt the chronic neurodegenerative process of AD prior to any loss of life quality.

AD patient cohorts in these studies include patients with a clinical diagnosis of dementia, specified as sporadic or genetic AD. Early onset familial AD is caused by rare, dominantly inherited mutations in amyloid precursor protein (APP) and presenilin proteins (*PSEN1* and *PSEN2*), and accounts for approximately 1% of all AD cases [117]. An interesting research avenue would be to compare the tau biomarker signature of patients with autosomal dominant familial AD to patients with sporadic AD.

Currently, only P-tau and T-tau are validated biomarkers for the AD continuum. The present authors believe that an individual’s “tauosome” signature, made up of a complex and diverse family of protein conformations, PTMs and fragments, could play an important role as a defining AD diagnostic biomarker. The translation of CNS-focussed tau biomarkers to the periphery is encouraging and could potentially pave the way for a more accessible primary screen to identify those at risk of becoming symptomatic.

The search for tau diagnostics in plasma is a rapidly expanding field with the potential to revolutionise the diagnosis of AD and other tauopathies. “Test, test, test” became the WHO and global mantra for overcoming the COVID-19 pandemic. A similar objective may contribute to beating back the relentless march of the already existing global AD epidemic. Removing the burden of this testing from specialist diagnostic units into a primary care setting may permit screening of far more people, giving us the best opportunity to preserve the healthspan of individual patients with disease modifying therapy.

## Figures and Tables

**Figure 1 ijms-21-08673-f001:**
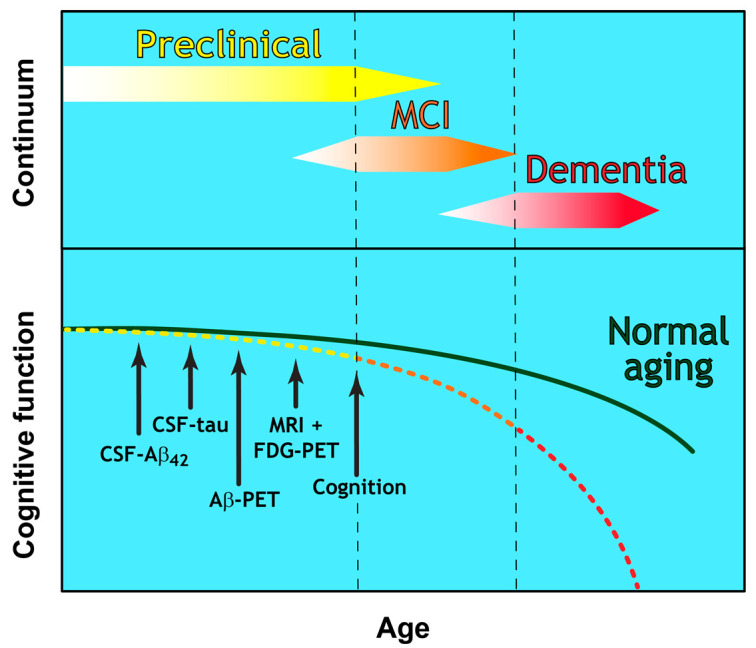
The Continuum of Alzheimer’s disease. The clinical trajectory of three defined and overlapping stages of Alzheimer’s disease (preclinical, Mild Cognitive Impairment (MCI), and dementia) in relation to the normal aging process. Cognitive function deteriorates with advancing age and at a greater extent than in normal aging. A hypothetical model showing the temporal sequence for the detection threshold of CSF and imaging biomarkers is indicated by arrows.

**Figure 2 ijms-21-08673-f002:**
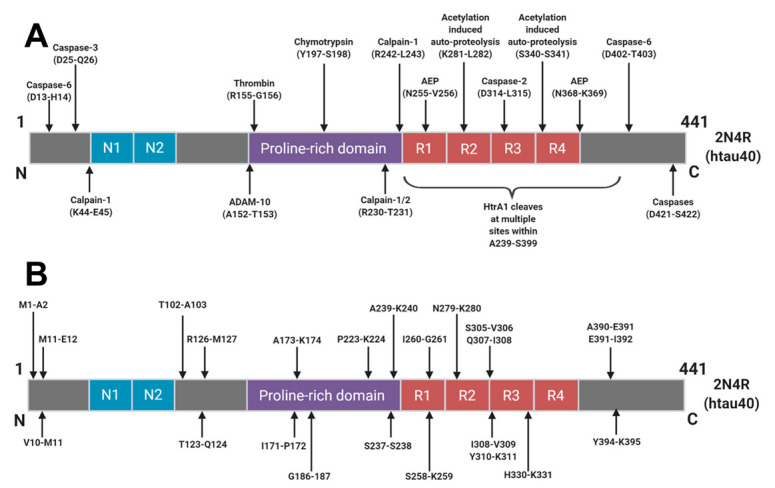
Schematic of Human Tau (2N4R) Cleavage Sites: Identified proteolytic cleavage sites from (**A**) those with known proteases responsible and (**B**) cleavage sites where the protease responsible has yet to be identified. Reprinted from [46]. Copyright (2018) IOS Press and authors under terms of CC BY-NC 4.0.

**Figure 3 ijms-21-08673-f003:**
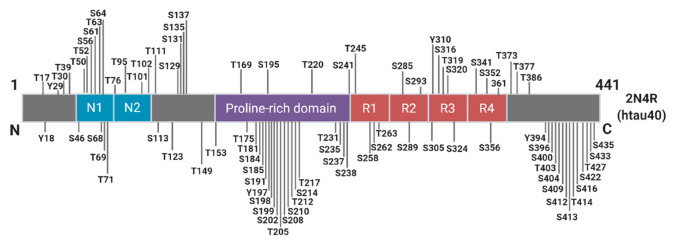
Schematic of Known Tau Phosphorylation Sites. Sites below the depicted 2N4R tau fragment have been identified in the Alzheimer’s brain. Information derived from [60].

**Table 1 ijms-21-08673-t001:** ATN Biomarker Classifications. From [11] with permission from Elsevier.

Classification	Definition	Biomarker
A	Aggregated Aβ or associated pathologic state	CSF Aβ42, or Aβ42/Aβ40 ratio
T	Aggregated tau or associated pathologic state	CSF Phosphorylated tau Tau PET
N	Neurodegeneration or neuronal injury	Anatomic MRI FDG PET CSF total tau

ATN Biomarker Grouping: Aβ—Amyloid β, CSF—Cerebrospinal fluid, FDG—Fluorodeoxyglucose, MRI—Magnetic resonance imaging, PET—Positron emission tomography.

**Table 2 ijms-21-08673-t002:** ATN biomarker profiles in relation to the clinical phase of Alzheimer’s Disease (AD) continuum. From [11], with permission from Elsevier.

Numeric Clinical Stage	Clinical Phase	ATN Classification
1	Cognitively normal with no indication of decline	A+ T− N−
2	Cognitively normal with indication of decline	A+ T+ N−
3	Prodromal AD	A+ T+ N+
4	Mild AD dementia	A+ T+ N+
5	Moderate AD dementia	A+ T+ N+
6	Severe AD dementia	A+ T+ N+

**Table 3 ijms-21-08673-t003:** ATN biomarker profiles in relation to the AD continuum. From [11] with permission from Elsevier.

A	T	N	Biomarker Category	
−	−	−	Normal	
+	−	−	Alzheimer’s pathologic change	AD Continuum
+	+	−	Alzheimer’s disease	
+	+	+	Alzheimer’s disease	
+	−	+	Alzheimer’s disease Concomitant non-Alzheimer’s pathologic change	
−	+	−	Non-AD pathologic change	
−	−	+	Non-AD pathologic change	
−	+	+	Non-AD pathologic change	

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
