# Peer review of "Current Progress and Future Directions for Tau-Based Fluid Biomarker Diagnostics in Alzheimer’s Disease"

_ijms, 2020, doi:10.3390/ijms21228673_

Round 1
Reviewer 1 Report
This manuscript provides very detailed and well organized information about tau used as a biomarker for Alzheimer’s disease. I was just missing reason for considering tau as a biomarker. The authors provide extensive information against the amyloid hypothesis but do not mention the evidence for involvement of tau in Alzheimer disease pathogenesis.
Minor points:
- All abbreviations should be explained, e.g., EMA and SIMOA
- Table captions should be placed above the tables and each table provided on a separate page
- Page 9, line 370: omit “a” in ….has a shown ….
Author Response
Dear Reviewer 1,
Firstly, many thanks for your constructive comments. Please find our commentary on your comments below in bold:
This manuscript provides very detailed and well organized information about tau used as a biomarker for Alzheimer’s disease. I was just missing reason for considering tau as a biomarker. The authors provide extensive information against the amyloid hypothesis but do not mention the evidence for involvement of tau in Alzheimer disease pathogenesis.
This is a fair point and the manuscript has been subsequently edited – see track changes
Minor points:
All abbreviations should be explained, e.g., EMA added and SIMOA introduced on page 9. All abbreviations should now be explained in text.
Table captions should be placed above the tables – In line with the journal template provided to us, we have kept this the same i.e. table/figure caption including a descriptive title as the the first line of the legend.
For example, Li, D.; Scarano, S.; Lisi, S.; Palladino, P.; Minunni, M. Real-Time Tau Protein Detection by Sandwich-Based Piezoelectric Biosensing: Exploring Tubulin as a Mass Enhancer. Sensors 2018, 18, 946. doi: 10.3390/s18040946
Each table provided on a separate page – In line with reviewer 2’s comments, we have chosen to simply increase the spacing. We would prefer that Figure 1, Table 1, Table 2 and Table 3 were all included on a single page as a visual summary page of the ATN System in relation to the continuum of AD.
Page 9, line 370: omit “a” in ….has a shown …. Removed
Many thanks,
Lewis K. Penny, Richard Lofthouse and Mohammad Arastoo,
Scottish Biologics Facility (SBF)
University of Aberdeen
Website: https://www.abdn.ac.uk/sbf/
Reviewer 2 Report
For the authors: the findings they mention do not distinguish between familiar hereditary cases and sporadic cases of Alzheimer's. Obviously, data from sporadic cases are more difficult to collect. However, the hereditary disease is thought to be due to mutations that alter the metabolism of APP and Amyloid-beta, which are the basis of the Amyloid-beta hypothesis of Alzheimer's.
Could you specify when the data mostly come from hereditary cases of Alzheimer's?
Minor comments: Could you increase the space between the tables?
Unfortunately, the therapy is still missing.
Author Response
Firstly, many thanks for your constructive comments. Please find our commentary on your comments below in bold:
Could you specify when the data mostly come from hereditary cases of Alzheimer's?
Manuscript has been updated with this as a discussion point – see concluding remarks
Minor comments: Could you increase the space between the tables? Agreed and spacing has been improved.
Unfortunately, the therapy is still missing. Agreed. As reflected in this manuscript, we believe early diagnostics gives the best chance to help investigate and validate disease modifying therapies.
Many thanks,
Lewis K. Penny, Richard Lofthouse and Mohammad Arastoo,
Scottish Biologics Facility (SBF)
University of Aberdeen
Website: https://www.abdn.ac.uk/sbf/